# The impact of prediabetes on preclinical atherosclerosis in general apparently healthy population: A cross-sectional study

Natalia Anna Zieleniewska[1,2], Jacek Jamiołkowski[1], Małgorzata Chlabicz[1,3],
Adam Łukasiewicz[4], Marlena Dubatówka[1], Marcin Kondraciuk[1], Paweł Sowa[1],
Irina Kowalska[5], Karol Adam Kamiński[1,2]*

1 Department of Population Medicine and Lifestyle Diseases Prevention, Medical University of Bialystok, Bialystok, Poland, 2 Department of Cardiology, University Hospital of Bialystok, Bialystok, Poland, 3 Department of Invasive Cardiology, University Hospital of Bialystok, Bialystok, Poland, 4 Department of Radiology, Medical University of Bialystok, Bialystok, Poland, 5 Department of Internal Medicine and Metabolic Diseases, Medical University of Bialystok, Bialystok, Poland

* fizklin@wp.pl

## Abstract

### Background

The hypothesis that not only diagnosed diabetes (DM), but also milder dysglycemia may affect the development of atherosclerosis still requires further study. In our population-based study, we aimed to evaluate the impact of prediabetic state on preclinical atherosclerosis and whether it may affect the cardiovascular risk (CVR) in the general population.

### Methods

The analysis was a part of the Bialystok PLUS cohort study and represented a random sample of Bialystok (Poland) residents aged 20–79 years at the time of sampling (July 2017-January 2023). The cross-sectional analysis included 1431 participants of a population-based study (mean age 46.82 years). Comprehensive biochemical assessments were performed. An Oral Glucose Tolerance Test (OGTT) was performed on fasting patients who did not report having a DM.

### Results

The population with prediabetes, based on HbA1c and OGTT, accounted for more than half of the study participants (n = 797, 55.7%). Atherosclerotic plaques in the carotid arteries were significantly more common in individuals with prediabetes considering all CVR categories. Prediabetes was associated with the occurrence of more advanced preclinical atherosclerosis, especially in the low to moderate CVR category. Serum glucose concentration after 1h and HbA1c proved to be statistically significant indicators of the presence of atherosclerotic plaques in ultrasound (respectively, AUC = 0.73 and 0.72). In multivariate logistic regression, prediabetes was independently associated with significantly increased risk of

**Data Availability Statement:** All relevant data are within the manuscript and its Supporting Information files.

**Funding:** The project was supported by statutory funds of the Medical University of Bialystok (B. SUB.23.172). The manuscript contains data acquired during the project VAMP financed by the National Centre for Research and Development (POIR.04.01.04-00-0052/18) The funders had no role in study design, data collection and analysis, decision to publish, or preparation of the manuscript.

**Competing interests:** The authors have declared that no competing interests exist.

**Abbreviations:** IFG, impaired fasting glucose; IGT, impaired glucose tolerance; DM, diabetes mellitus; ASCVD, atherosclerotic cardiovascular disease; TC, total cholesterol; LDL-C, LDL-cholesterol; HDL-C, HDL-cholesterol; TG, triglycerides; HbA1c, glycated hemoglobin; WHO, World Health Organization; OGTT, Oral Glucose Tolerance Test; Cobas c111, Roche, reference enzymatic method with hexokinase; HPLC, ion-exchange high performance liquid chromatography; ECLIA, electrochemiluminescence; BMI, body mass index; WHR, Waist-to-hip ratio; BP, blood pressure; CCA, common carotid artery; ECA, external carotid artery; ICA, internal carotid artery; BIF, bifurcation; IMT, intima media thickness; PWV, pulse wave velocity; AIx, augmentation index; CP, central pressure; SCORE2, Systematic Coronary Risk Estimation 2; SCORE2-OP, Systematic Coronary Risk Estimation 2-Older Persons; CVD, cardiovascular disease; MI, myocardial infarction; IHD, ischemic heart disease; TIA, transient ischemic attack; PAD, peripheral arterial disease; CKD, chronic kidney disease; ESC, European Society of Cardiology; EAS, European Atherosclerosis Society; ACR, albumin-to-creatinine ratio; eGFR, estimated glomerular filtration rate; HOMA-IR, Homeostasis model assessment of insulin resistance; QUICKI, Quantitative insulin sensitivity check index; EASD, European Association for the Study of Diabetes; AUC, area under the curve; US, ultrasound; ACS, acute coronary syndrome; CI, Confidence interval.

preclinical atherosclerosis (OR = 1.56, 95% CI 1.09–2.24), along with CVR categories, pulse wave velocity and central blood pressure augmentation index.

## Conclusions

Prediabetes is associated with the occurrence and progression of the preclinical atherosclerosis. Importantly, many of those patients are in the low to moderate cardiovascular risk category, hence may have a severely underestimated risk. Inclusion of prediabetes into CVR assessment may improve risk stratification. An early identification of dysglycemic population is necessary to effectively implement the cardiovascular and metabolic prevention measures.

## Introduction

Prediabetes is an intermediate state defined as raised blood glycemic parameters above the normal range, but below the threshold for diabetes [1]. Prediabetes is specifically defined as impaired fasting glucose (IFG) or impaired glucose tolerance (IGT) [1]. Such a condition is classified as dysglycemia and a prelude to diabetes mellitus (DM) [2]. The incidence of type 2 DM progression five years after the diagnosis of IGT or IFG is estimated at 26% and 50%, respectively [2]. The natural progression of prediabetes is increasing insulin resistance and pancreatic B-cell dysfunction, leading to overt DM [3]. The prevalence of non-diabetic hyperglycaemia is steadily increasing: 541 million adults (10.6% of adults worldwide) are estimated to have IGT and 319 million adults (6.2%) to have IFG [4].

Early diagnosis of prediabetes is crucial from clinical standpoint. First, the presence of non-diabetic hyperglycemia signifies an increased risk of developing type 2 DM [5, 6], which accounts for over 90% of all diabetes worldwide [4]. Prediabetes identifies an enhanced incidence of cardiovascular disease [7, 8]. Importantly, early diagnosis of prediabetes opens up opportunities for therapeutic interventions to prevent progression to DM [9]. However, the difficulty in diagnosing hyperglycemia is that it often remains asymptomatic. Symptoms, such as excessive thirst, frequent urination and fatigue, often appear very late in the course of the disease and may be ignored by the patient or considered insignificant. As a result, even half of people with dysglycemia in the population may be undiagnosed [4]. There is marked diagnostic inertia both in diagnosing and treating prediabetes, therefore it is of utmost clinical importance to present and advocate the unfavourable consequences of dysglycemia. Importantly many of these people consider themselves healthy and their increased risk remains unrecognized by healthcare providers, as prediabetes is not included in the major cardiovascular risk scores.

Obesity, hypertension, cigarette smoking, and DM are well-recognized risk factors for atherosclerotic cardiovascular disease (ASCVD) [10]. ASCVD remains the most widespread cardiovascular problem, moreover, myocardial infarction is considered to be the leading cause of death [1]. A previously conducted study of patients with first-time acute coronary syndrome (ACS) who underwent urgent coronary angiography showed that coronary atherosclerosis is more advanced in patients with prediabetes than in patients without DM [38]. Furthermore, the association of the occurrence of coronary atherosclerosis was shown to be comparable between patients with prediabetes and patients with DM at the time of first ACS [38]. Another study using percutaneous coronary intervention showed that coronary lesions in prediabetic patients were associated with higher levels of lipid-rich atherosclerotic plaques [39].

Prediabetes is associated with inflammation and vasoconstriction, which promote atherosclerosis in the coronary arteries [40]. Notably, limited evidence comes from research on both, prediabetes and carotid atherosclerosis in general population.

Recently, cardiovascular prevention and identifying people at risk for ASCVD as early as possible have grown in importance. The aim of our study was to analyse whether prediabetes affects the development of preclinical atherosclerosis in the general population.

## Material and methods

### Cohort study Bialystok PLUS

Our analysis was a part of the Bialystok PLUS cohort study and represented a random sample of Bialystok residents aged 20–79 years at the time of sampling (July 2017-January 2023). Bialystok is a medium-sized city located in eastern Poland with a population of 293,400. The recruitment of participants for the population-based study uses a pseudonymized list of residents of Bialystok obtained from the Local Municipal Office. Annually, we randomly sampled citizens to obtain a distribution of proportions in terms of age and gender similar to that of the city's population. A more detailed study design was described in previously published paper [11]. On June 1, 2023, the data was made available for research purposes. The study authors did not have access to information that could identify individual participants after data collection.

The study was conducted in accordance with the Declaration of Helsinki. Written informed consent was obtained from all participants. Ethical approval for this study was provided by the Ethics Committee of the Medical University of Bialystok (Poland) on 31 March 2016 (approval number: R-I-002/108/2016).

### Data collection and assays

All the clinical and biochemical measurements were conducted by qualified medical personnel. High reproducibility of the examinations was achieved by performing them based on verified Standard Operating Procedures. The details of the subjects' medical history were collected from questionnaires at the time of study entry. In the morning of the visit, peripheral intravenous fasting blood samples were collected after at least eight hours of fasting. Patients also declared that they slept through the night. Comprehensive biochemical assessments were performed: total cholesterol (TC), LDL-cholesterol (LDL-C) and HDL-cholesterol (HDL-C) fractions, triglycerides (TG), glucose, glycated hemoglobin (HbA1c). The samples were then prepared for further analysis by centrifugation and storage at −70°C. As recommended by the World Health Organization (WHO), we performed an Oral Glucose Tolerance Test (OGTT) on fasting patients who did not report having a history of DM. 75 grams of glucose dissolved in water was administered and then blood was taken sequentially after 1 and 2 hours.

Fasting glucose concentration at 60 min and 120 min glucose levels were assessed in plasma drawn on EDTA with sodium fluoride using a reference enzymatic method with hexokinase (Cobas c111, Roche). Using the homogeneous enzymatic colorimetric method on the Cobas c111 from ROCHE (ROCHE, Meylan, Isère, France), the concentration of low- and high-density lipoprotein cholesterols (LDL-C and HDL-C, respectively) were determined. Total cholesterol and triglycerides were determined using the enzymatic colorimetric method on the Cobas c111 from ROCHE. Glycated hemoglobin A1c (HbA1c) was determined by ion-exchange high performance liquid chromatography (HPLC) on D-10 from Bio-Rad (Bio-Rad, Hercules, CA, USA). Serum insulin was determined by electrochemiluminescence (ECLIA) on the Cobas e411 device (ROCHE).

Anthropometric measurements including height and circumferences of the waist, abdomen and hips were taken (SECA 201 tape, Hamburg, Germany). Body mass index (BMI) was calculated as weight in kilograms divided by height in metres squared and is expressed in units of kg/m$^2$. Waist-to-hip ratio (WHR) was calculated as a ratio between waist and hips circumference. Blood pressure (BP) was measured using the oscillometric method (obtained with the Omron Healthcare Co. Ltd MG Comfort device) after the participants were seated for at least 5 minutes—two measurements were taken 5 minutes apart and the average was drawn. The weight and body mass composition were determined using the InBody 770.

Ultrasound examination (US) of the carotid arteries was used to evaluate early atherosclerotic lesions. The ultrasonography measurements were made using the ultrasound Vivid 9 (GE Healthcare, Chicago, IL, USA). The presence of any atherosclerotic plaques in 1) right common carotid artery (CCA), 2) left CCA, 3) right external carotid artery (ECA), 4) left ECA, 5) right internal carotid artery (ICA), 6) left ICA, 7) right bifurcation (BIF), 8) left BIF were evaluated. We assessed atherosclerotic plaques as binomial quality variables and marked them as present when 1 of the following criteria was fulfilled: 1) the local thickening IMT towards the lumen of the vessel, exceeding the surrounding IMT by > 0.5 mm, 2) the local thickening IMT towards the lumen of the vessel, surpassing the surrounding IMT by 50%, 3) IMT thickening > 1.5 mm [12]. A constant element of the study was the estimation of intima media thickness (IMT) in right and left CCA and the result is presented as an average value from 5 measurements. In addition, we assessed the advanced stage of early atherosclerotic lesions based on the presence of ICA stenosis. Stenosis severity was determined using the North American Symptomatic Carotid Endarterectomy Trial (NASCET) criteria [13].

The parameters for assessing arterial stiffness, i.e. carotid-femoral pulse wave velocity (PWV) and augmentation index (AIx), were measured using an oscillometric method (Vascular Explorer, Enverdis, Jena, Germany) in the supine position preceded by 10 minutes of rest.

## Division into cardiovascular risk classes

To calculate CV risk in primary prevention were used Systematic Coronary Risk Estimation 2 (SCORE2) and Systematic Coronary Risk Estimation 2-Older Persons (SCORE2-OP) [10]. The SCORE2 and SCORE2-OP estimates an individual's 10-year risk of fatal and non-fatal CVD events (myocardial infarction, stroke) in apparently healthy people. The SCORE2 and SCORE2-OP was calculated, excluding participants who were pre-qualified in the high and very high CV risk classes, i.e. participants with previously diagnosed CVD (myocardial infarction—MI, ischemic heart disease—IHD, stroke, transient ischemic attack—TIA, peripheral arterial disease—PAD, significant plaque on carotid ultrasound >50% of arterial stenosis), moderate or severe chronic kidney disease (CKD) at the time of study entry, and younger than 40 years old. The SCORE2 was calculated for those aged 40–69, while SCORE2-OP was calculated for those aged 70–89. The calculator for high CVD risk countries was used, as Poland belongs to this category.

The study population was divided according to the latest recommendation "2021 ESC Guidelines on cardiovascular disease prevention in clinical practice" [10] into low-to-moderate, high and very-high CVD categories. Firstly, high and very-high risk individuals were identified. The previously calculated SCORE2 and SCORE2-OP values were then used to categorize apparently healthy individuals.

Subjects with familial hypercholesterolemia, DM type 1 and 2 and with established ASCVD were assigned to appropriate categories according to "2021 ESC Guidelines on cardiovascular disease prevention in clinical practice" [10]. Individuals with CKD were assigned to high and very-high CVD risk categories according to previous guidelines: "2019 ESC/EAS guidelines

for the management of dyslipidemias: lipid modification to reduce cardiovascular risk" [14] due to lacking the albumin-to-creatinine ratio (ACR) in the studied population. To high CVD risk category were classified people with estimated glomerular filtration rate (eGFR) 30-59mL/min/1.73m$^2$, while to very high CVD risk category with eGFR <30mL/min/1.73m$^2$.

### Definition of diabetes, prediabetes and assessment of insulin sensitivity

We used the diagnostic criteria for DM and prediabetes according to ESC and the European Association for the Study of Diabetes (EASD) recommendations [1]. DM was not diagnosed based on fasting glucose due to availability of a single measurement. Prediabetes was diagnosed in participants who had no previous diabetes: 1) IFG if fasting glycaemia was between 100–125 mg/dl (5.6–6.9 mmol/l) and after 2h was less than 140 mg/dl (7.8 mmol/l); 2) IGT if at 2h glycaemia was ≥140 mg/dl (7.8 mmol/l) and < 200 mg/dl (11.1 mmol/l); 3) based on HbA1c when it ranged from 5.7–6.4% (42–47 mmol/mol). Those who had fasting glucose in a single measurement above 125 mg/dl and after 2h below 200 mg/dl were categorised as IGTs. Newly diagnosed DM was diagnosed based on OGTT ≥ 200 mg/dl and an HbA1c ≥ 6.5%. We used several well-described indicators to assess insulin resistance. Quantitative insulin sensitivity check index (QUICKI) was defined as $1/[\log(I_0) + \log(G_0)]$, where $I_0$ is the fasting insulin (μU/ml), and $G_0$ is the fasting glucose (mmol/l) [15]. Homeostasis model assessment of insulin resistance (HOMA-IR) was calculated using the formula: $I_0 \times G_0 / 22.5$ [16]. When assessing insulin sensitivity, we used the Matsuda index calculated from the formula $10000/\sqrt{(G_0 \text{ mg/dl} \times I_0 \text{ μU/ml})} \times$ (Mean OGTT glucose concentration mg/dl × mean OGTT insulin concentration mg/dl) [17].

### Statistical analysis

Descriptive statistics for continuous variables were presented as means and 95% Confidence Intervals for means and for categorical variables as proportions and 95% Confidence Intervals for proportions. The sample size is large enough to assume that the variables followed a normal distribution according to Central Limit Theorem. Comparisons of variables between subgroups were analysed using the chi-squared test for qualitative variables. Two regression logistic models were performed. In one model, the independent variables were classic well-described cardiovascular risk factors and in the other one we used stratification into cardiovascular risk classes (which includes few of these classic risk factors). Regression models were presented using regression coefficients, p-value of Wald tests and coefficients of determination for the model. The IBM SPSS Statistics 27.0 statistical software (Armonk, NY, USA) was used for all the calculations. The criterion for statistical significance was set at $p < 0.05$. The figures have been made in Excel and R 4.2.1.

### Results

### Study population

The study enrolled 1866 participants. According to STROBE guidelines, the flowchart represents the study population (Fig 1). Prevalent diabetes (n = 252, 13.5%) was diagnosed on the basis of medical history (n = 140) and glucose levels after 2h OGGT (n = 112). All prevalent diabetic patients were excluded from further analysis. Additionally, 183 patients were excluded from further analysis for technical reasons–inability to perform an OGTT (n = 132) or US of the carotid arteries (n = 4) or lack of the HbA1c measurement (n = 7) or lack of CVR assessment (n = 40) [18]. After applying these exclusion criteria, the data of 1431 participants (mean age 46.82 years) remained for analysis. The population of attendees without dysglycemia

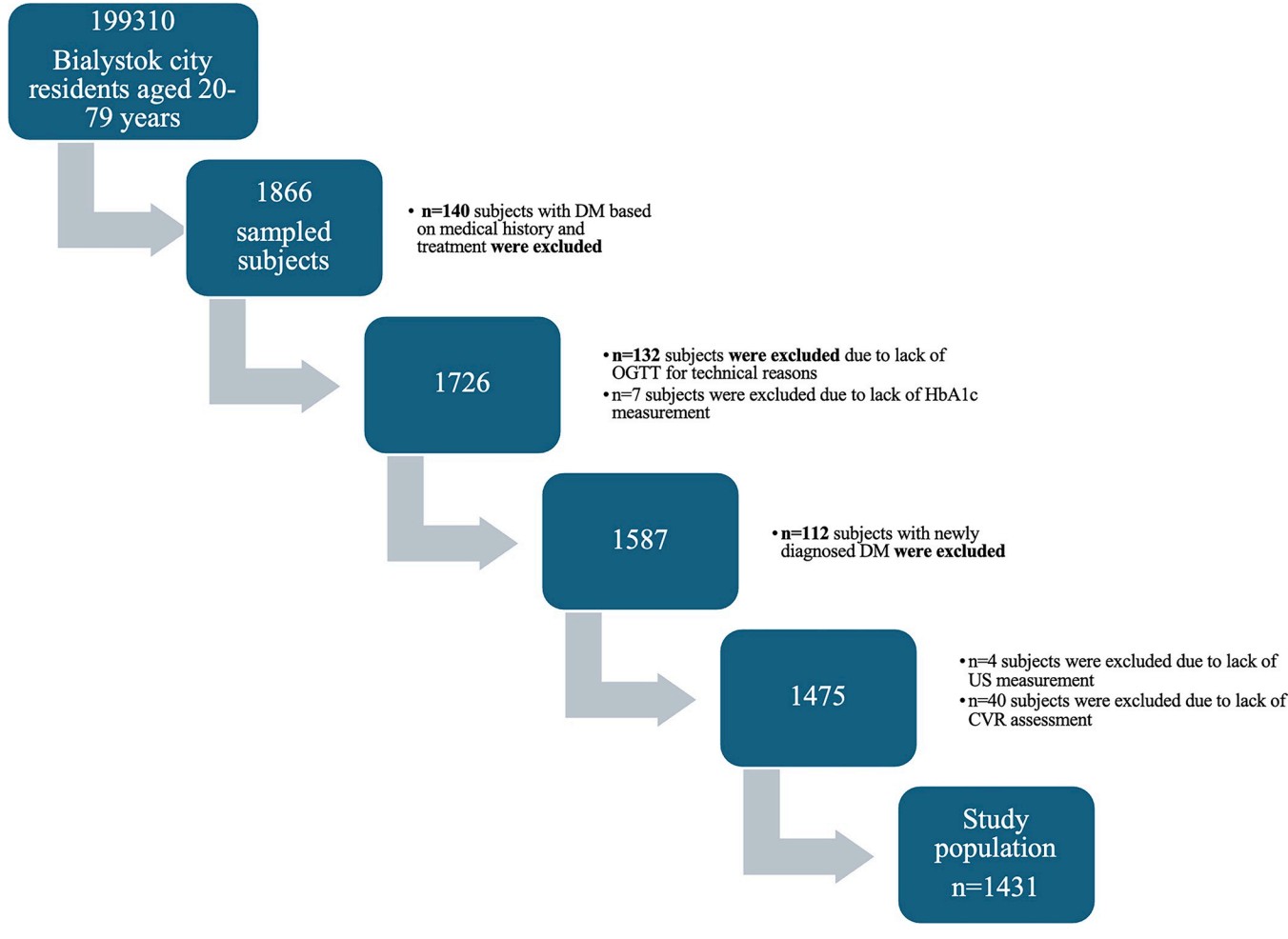

**Fig 1. Flowchart of study population.**

consisted of 634 patients (44.3%). The population with prediabetes accounted for more than half of the study participants (n = 797, 55.7%). Subsequently, we outlined the differences in the diagnosis of prediabetes considering the diagnostic method. It is worth noting that as many as 84 (approximately 5.9%) patients with normal OGTT results had elevated hemoglobin HbA1c values.

Table 1 presents the basic characteristics of the study group by glucose metabolism status. The population without glucose metabolism disturbances was significantly different in all parameters describing metabolic and anthropometric values and preclinical atherosclerosis comparing to that of prediabetes. Patients with normal glycemia (both fasting and OGTT), but with elevated HbA1c values significantly differed from patients without impaired glucose metabolism in the following parameters: age, diastolic blood pressure, LDL-C, QUICKI, glucose levels after 1h, mean CCA IMT, as well as the presence of atherosclerotic plaques and arterial stiffness parameters. Moreover, we found no differences in anthropometric parameters and body composition. We also noted that there was a statistically significant difference in the presence of atherosclerotic plaques between the group with IFG and those with IGT.

To estimate possible predictors of the atherosclerotic plaques' formation (dependent variable), ROC curves for glycemic parameters and insulin resistance indices were created (S1 Fig, S1 Table). We observed that all examined parameters proved to be statistically significant

**Table 1. Baseline clinical and metabolic characteristics of population group and groups with prediabetes.**

| | Total n = 1431 | No glucose metabolism disturbances n = 634 Mean (95% CI) | Population with prediabetes n = 797 | | |
| --- | --- | --- | --- | --- | --- |
| | | | IFG n = 415 Mean (95% CI) | IGT n = 298 Mean (95% CI) | Population with prediabetes based on HbA1c alone n = 84 Mean (95% CI) |
| Age, years | 46.82 (46.05–47.59) | 39.60 (38.62–40.57) | 50.15 (48.83–51.46) | 55.46 (53.87–57.05) | 54.23 (50.99–57.47) |
| Sex, male, n, % | 52.8% (43.4%-62.1%) | 38.8% (35.1%-42.7%) | 57.8% (53.0%-62.5%) | 45.3% (39.7%-51.0%) | 26.2% (18.0%-36.5%) |
| BMI, kg/m$^2$ | 26.30 (26.06–26.54) | 24.70 (24.39–25.02) | 27.48 (27.06–27.90) | 28.21 (27.67–28.76) | 25.78 (24.85–26.71) |
| WHR | 0.86 (0.85–0.86) | 0.83 (0.82–0.83) | 0.89 (0.88–0.90) | 0.89 (0.88–0.90) | 0.84 (0.83–0.86) |
| SBP mmHg | 122.18 (121.31–123.04) | 116.63 (115.51–117.76) | 126.91 (125.33–128.49) | 128.03 (125.97–130.08) | 119.93 (116.81–123.06) |
| DBP mmHg | 80.48 (79.96–80.99) | 78.04 (77.36–78.72) | 82.79 (81.82–83.77) | 82.98 (81.76–84.20) | 78.60 (76.44–80.77) |
| Total cholesterol, mg/dl, | 193.63 (191.59–195.67) | 185.15 (182.46–187.84) | 202.46 (198.47–206.45) | 195.97 (191.36–200.58) | 205.70 (196.20–215.21) |
| Triglycerides, mg/dl | 109.45 (105.34–113.56) | 96.59 (89.84–103.34) | 116.00 (109.74–122.25) | 128.71 (119.21–138.21) | 105.86 (95.64–116.08) |
| HDL-C, mg/dl, | 62.17 (61.31–63.03) | 63.83 (62.58–65.09) | 60.91 (59.23–62.59) | 59.47 (57.72–61.21) | 65.42 (61.63–69.21) |
| LDL-C, mg/dl, | 123.86 (122.00–125.72) | 115.68 (113.23–118.13) | 132.76 (129.14–136.38) | 126.15 (122.03–130.26) | 133.36 (124.28–142.45) |
| Cholesterol-lowering treatment, % | 25.5% (18.1%-34.5%) | 2.7% (1.7%-4.3%) | 10.8% (8.2%-14.2%) | 20.5% (16.3%-25.4%) | 17.9% (11.1%-27.4%) |
| Fasting glucose level, mg/dl | 97.87 (97.38–98.36) | 90.78 (90.34–91.22) | 105.77 (105.25–106.28) | 103.40 (102.38–104.41) | 92.70 (91.60–93.80) |
| Serum glucose concentration, 1h after glucose loading, mg/dl | 143.57 (141.33–145.82) | 120.85 (118.29–123.40) | 152.38 (148.91–155.85) | 178.48 (173.99–182.97) | 133.75 (124.61–142.89) |
| Serum glucose concentration, 2h of glucose loading, mg/dl, | 117.70 (116.24–119.17) | 102.43 (100.99–103.87) | 112.54 (110.70–114.37) | 158.98 (157.20–160.75) | 112.08 (107.88–116.29) |
| Fasting insulin concentration, mg/dl, | 11.81 (11.41–12.21) | 9.63 (9.23–10.04) | 13.45 (12.71–14.20) | 14.92 (13.71–16.14) | 9.94 (8.96–10.91) |
| Serum insulin concentration, 1h after glucose loading, mg/dl, | 84.87 (81.09–88.65) | 66.26 (62.43–70.09) | 98.36 (90.74–105.98) | 105.73 (95.33–116.12) | 77.68 (66.20–89.16) |
| Serum insulin concentration, 2h after glucose loading, mg/dl, | 61.99 (58.78–65.21) | 43.38 (40.99–45.77) | 56.55 (52.03–61.07) | 110.48 (99.53–121.42) | 55.82 (47.75–63.88) |
| HbA1c, % | 5.36 (5.34–5.38) | 5.16 (5.14–5.18) | 5.45 (5.42–5.48) | 5.51 (5.47–5.55) | 5.84 (5.81–5.86) |
| HOMA-IR | 2.90 (2.80–3.01) | 2.17 (2.08–2.27) | 3.52 (3.32–3.72) | 3.87 (3.54–4.20) | 2.29 (2.06–2.52) |
| QUICKI | 0.34 (0.33–0.34) | 0.35 (0.35–0.35) | 0.32 (0.32–0.33) | 0.32 (0.32–0.33) | 0.34 (0.34–0.35) |
| Matsuda Index | 4.69 (4.52–4.85) | 6.04 (5.77–6.31) | 3.79 (3.57–4.01) | 3.05 (2.82–3.28) | 5.29 (4.46–6.12) |
| Average CCA IMT | 0.63 (0.62–0.64) | 0.58 (0.57–0.59) | 0.65 (0.64–0.66) | 0.70 (0.68–0.71) | 0.67 (0.64–0.69) |
| Presence of any atherosclerotic plaques | 70.8% (61.5%-78.6%) | 23.3% (20.2%-26.8%) | 50.4% (45.6%-55.1%) | 61.4% (55.8%-66.8%) | 50.0% (39.5%-60.5%) |

*(Continued)*

**Table 1.** (Continued)

| | Total n = 1431 | No glucose metabolism disturbances n = 634 Mean (95% CI) | Population with prediabetes n = 797 | | |
| --- | --- | --- | --- | --- | --- |
| | | | IFG n = 415 Mean (95% CI) | IGT n = 298 Mean (95% CI) | Population with prediabetes based on HbA1c alone n = 84 Mean (95% CI) |
| Brachial- Ankle PWV, m/s | 10.36 (10.24–10.49) | 9.58 (9.42–9.74) | 10.72 (10.51–10.93) | 11.30 (11.03–11.57) | 10.72 (9.95–11.50) |
| AIx | 14.71 (14.06–15.36) | 12.20 (11.28–13.11) | 15.40 (14.20–16.60) | 17.86 (16.51–19.20) | 18.22 (14.88–21.56) |

BMI–Body Mass Index, WHR–Waist-Hip Ratio, SBP–systolic blood pressure, DBP–diastolic blood pressure, HbA1c –glycated hemoglobin A1c, LDL -low density lipoprotein, HDL–high density lipoprotein, HOMA-IR—Homeostasis Model Assessment of Insulin Resistance, QUICKI- quantitative insulin sensitivity check index, CCA IMT—Carotid Artery Intima Media Thickness, PWV–pulse wave velocity, Aix- augmentation index, CI–Confidence interval.

indicators of the presence of atherosclerotic plaques in carotid ultrasound, also upon application of Bonferroni correction. However, the highest AUC values were obtained for serum glucose concentration after 1h and HbA1c (AUC = 0.73; 95% CI 0.70–0.76 for glucose level after 1 h and AUC = 0.72; 95% CI 0.69–0.75 for HbA1c).

Atherosclerotic plaque presence in particular cardiovascular risk categories.

Fig 2 presented the prevalence of prediabetes in the general population with cardiovascular risk categories and the presence of atherosclerotic plaques in the carotid arteries. The majority, two-thirds of the study population were patients classified in CVR category 1. Moreover, prediabetes was associated with the occurrence of preclinical atherosclerosis in the low to moderate CVR category (p<0.001, S2 Table). Atherosclerotic plaques in the carotid arteries were significantly more common in hyperglycemic individuals considering all CVR categories (p<0.001, S3 Table). In the next tables, we showed that more advanced preclinical atherosclerotic lesions are statistically more common in prediabetic patients (p<0.001; S4 and S5 Tables).

In the Table 2, we have presented the characteristics of the study group by the presence or absence of atherosclerotic lesions and their severity. In all three conditions, individuals with the presence of advanced atherosclerotic lesions were significantly older than those without. As expected, subjects with and without any atherosclerotic plaques differed both in lipid profile, glycemic and insulin resistance tests and also in anthropometric measurements. Observing more advanced lesions (ICA stenosis >50%), we could see that the groups do not vary in lipid profile or BMI, although we may observed significant differences in glycemic and glycated hemoglobin measurements and cholesterol-lowering treatment. Moreover, patients with any of the three conditions had significantly higher arterial stiffness parameters.

We also performed multivariate logistic regression model where the dependent variable was the presence of any atherosclerotic plaques (Table 3). We built a model in which we included CVR categories (which already include classic cardiovascular risk factors, including gender and age), glucose metabolism and vascular stiffness parameters. Prediabetes was associated with significantly increased risk of preclinical atherosclerosis (OR = 1.56, 95% CI 1.09–2.24; p = 0.014), along with CVR categories, pulse wave velocity and central blood pressure augmentation index. Based on these results, we conclude that prediabetes was independently associated with preclinical atherosclerosis. This association remained significant after adjustment for confounders and the final findings were shown in Table 3.

## Discussion

Our findings indicate an association between prediabetes and preclinical atherosclerosis based on the presence of plaques in the carotid arteries, both in the general population and in the

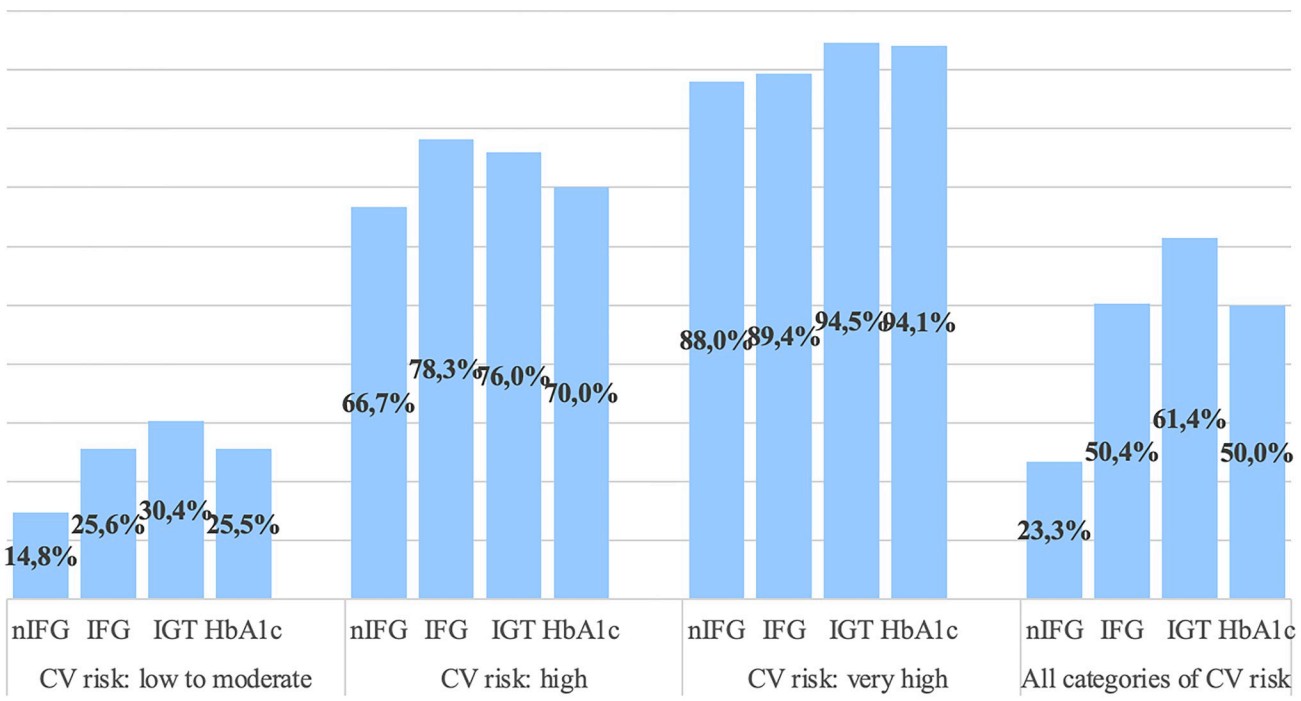

IFG - impaired fasting glucose, IGT - impaired glucose tolerance, HbA1c - glycated hemaglobin A1c, nIFG – population without impaired glucose metabolism.

**Fig 2. Prevalence of prediabetes in the general population with cardiovascular risk categories and the presence of atherosclerotic plaques in the carotid arteries.**

low-to-moderate CVR category. This is particularly important considering that up to 2/3 of all examined participants were categorized in this category. Patients with a low-to-moderate CVR may have an underestimated cardiovascular risk, due to impaired glucose metabolism, which is simply not considered in risk calculators. In a recent meta-analysis, researchers demonstrated that nondiabetic hyperglycaemia is associated with an increased risk of all-cause mortality and cardiovascular disease in the general population and in patients with atherosclerotic CVD [19]. A Norwegian population-based study showed that DM and prediabetic patients, compared to normoglycemic patients, had statistically significantly more often elevated CVD risk factors: higher BMI, higher waist circumference, lower HDL-C and higher CRP, besides being more likely to take lipid-lowering and blood pressure normalizing medications [20]. Our study also indicates significantly elevated ASCVD risk factors in patients with prediabetes compared to population without impaired glucose metabolism. Another meta-analysis found that people with prediabetes have a roughly 20% increased risk of CVD, regardless of its type (IFG vs IGT) [21]. Another large study revealed that both elevated fasting blood glucose and after 2h on the OGTT were associated with a higher risk of ASCVD, but with slightly higher relative effects of 2-hour glucose than fasting glucose [22]. Moreover, duration of prediabetes during adulthood is independently associated with subclinical atherosclerosis as shown in Coronary Artery Risk Development in Young Adults Study [23]. These data point out that early identification of patients with prediabetes and assessment of CVR factors may be important in early implementation of non- and pharmacological interventions and prevent the development of early atherosclerotic lesions. Recently published results indicated that prediabetes

**Table 2. Evaluation of the relationship of vascular early atherosclerotic parameters with anthropometric, metabolic and insulin resistance parameters.**

| | The occurrence of any atherosclerotic plaques in the carotid arteries | | ICA stenosis >50% | | Local IMT thickening | |
|---|---|---|---|---|---|---|
| | Mean (95% Confidence interval) | | Mean (95% Confidence interval) | | Mean (95% Confidence interval) | |
| Age | without any plaques | 38.54 (37.80–39.27) | Without stenosis | 46.56 (45.79–47.33) | Without IMT thickening | 38.59 (37.79–39.38) |
| | with plaques | 58.90 (57.98–59.81) | With ICA stenosis >50% | 69.93 (65.85–74.01) | With IMT thickening | 58.17 (57.15–59.18) |
| BMI kg/m2 | without any plaques | 25.28 (24.99–25.57) | Without stenosis | 26.28 (26.04–26.52) | Without IMT thickening | 25.34 (25.03–25.66) |
| | with plaques | 27.79 (27.42–28.17) | With ICA stenosis >50% | 28.17 (26.08–30.26) | With IMT thickening | 27.72 (27.33–28.11) |
| WHR | without any plaques | 0.83 (0.83–0.84) | Without stenosis | 0.86 (0.85–0.86) | Without IMT thickening | 0.83 (0.83–0.84) |
| | with plaques | 0.89 (0.89–0.90) | With ICA stenosis >50% | 0.92 (0.88–0.97) | With IMT thickening | 0.90 (0.89–0.90) |
| LDL-C mg/dl | without any plaques | 117.74 (115.55–119.93) | Without stenosis | 123.91 (122.05–125.78) | Without IMT thickening | 118.24 (115.81–120.67) |
| | with plaques | 132.80 (129.66–135.93) | With ICA stenosis >50% | 116.08 (93.80–138.37) | With IMT thickening | 131.45 (128.23–134.68) |
| Fasting glucose level, mg/dl | without any plaques | 95.48 (94.88–96.08) | Without stenosis | 97.77 (97.28–98.27) | Without IMT thickening | 96.37 (95.72–97.01) |
| | with plaques | 101.35 (100.61–102.08) | With ICA stenosis >50% | 105.93 (102.08–109.77) | With IMT thickening | 101.84 (101.11–102.57) |
| Serum glucose concentration after 1h in OGTT, mg/dl | without any plaques | 132.08 (129.43–134.74) | Without stenosis | 143.12 (140.87–145.37) | Without IMT thickening | 132.13 (129.41–134.85) |
| | with plaques | 160.18 (156.74–163.62) | With ICA stenosis >50% | 179.50 (163.25–195.75) | With IMT thickening | 159.19 (155.82–162.56) |
| Serum glucose concentration after 2h in OGTT, mg/dl | without any plaques | 111.67 (109.91–113.43) | Without stenosis | 117.57 (116.10–119.05) | Without IMT thickening | 112.93 (111.03–114.83) |
| | with plaques | 126.50 (124.16–128.85) | With ICA stenosis >50% | 126.07 (113.90–138.25) | With IMT thickening | 126.24 (123.86–128.62) |
| HbA1c % | without any plaques | 5.24 (5.22–5.27) | Without stenosis | 5.35 (5.33–5.37) | Without IMT thickening | 5.24 (5.22–5.27) |
| | with plaques | 5.52 (5.49–5.55) | With ICA stenosis >50% | 5.90 (5.72–6.08) | With IMT thickening | 5.51 (5.48–5.54) |
| Fasting insulin concentration, μU/ml | without any plaques | 10.92 (10.47–11.37) | Without stenosis | 11.77 (11.37–12.18) | Without IMT thickening | 11.11 (10.61–11.61) |
| | with plaques | 13.09 (12.38–13.81) | With ICA stenosis >50% | 13.45 (10.07–16.83) | With IMT thickening | 13.23 (12.46–14.00) |
| Serum insulin concentration after 1h in OGTT, μU/ml | without any plaques | 75.74 (71.81–79.67) | Without stenosis | 84.37 (80.58–88.17) | Without IMT thickening | 75.38 (71.43–79.34) |
| | with plaques | 97.97 (90.86–105.08) | With ICA stenosis >50% | 126.66 (90.83–162.48) | With IMT thickening | 97.22 (90.27–104.17) |
| Serum insulin concentration after 2h in OGTT, μU/ml | without any plaques | 52.79 (49.74–55.83) | Without stenosis | 61.75 (58.52–64.98) | Without IMT thickening | 53.96 (50.31–57.61) |
| | with plaques | 75.29 (68.94–81.63) | With ICA stenosis >50% | 76.92 (47.47–106.37) | With IMT thickening | 74.08 (67.56–80.61) |
| HOMA-IR | without any plaques | 2.61 (2.49–2.73) | Without stenosis | 2.89 (2.79–3.00) | Without IMT thickening | 2.69 (2.55–2.82) |
| | with plaques | 3.33 (3.13–3.52) | With ICA stenosis >50% | 3.52 (2.63–4.41) | With IMT thickening | 3.38 (3.17–3.59) |

*(Continued)*

**Table 2.** (Continued)

| | | The occurrence of any atherosclerotic plaques in the carotid arteries | | ICA stenosis >50% | | Local IMT thickening | |
|---|---|---|---|---|---|---|---|
| | | | | Mean (95% Confidence interval) | | Mean (95% Confidence interval) | |
| | | Mean (95% Confidence interval) | | | | | |
| QUICKI | without any plaques | 0.34 (0.34–0.34) | Without stenosis | 0.34 (0.33–0.34) | Without IMT thickening | 0.34 (0.34–0.34) | |
| | with plaques | 0.33 (0.33–0.33) | With ICA stenosis >50% | 0.32 (0.31–0.34) | With IMT thickening | 0.33 (0.33–0.33) | |
| Matsuda Index | without any plaques | 5.21 (4.98–5.44) | Without stenosis | 4.71 (4.54–4.87) | Without IMT thickening | 5.23 (5.00–5.46) | |
| | with plaques | 3.94 (3.71–4.16) | With ICA stenosis >50% | 3.11 (1.65–4.58) | With IMT thickening | 3.96 (3.74–4.18) | |
| Carotid-femoral PWV, m/s | without any plaques | 8.13 (7.99–8.26) | Without stenosis | 8.57 (8.44–8.69) | Without IMT thickening | 8.14 (8.00–8.27) | |
| | with plaques | 9.23 (9.00–9.45) | With ICA stenosis >50% | 8.75 (6.87–10.62) | With IMT thickening | 9.17 (8.95–9.39) | |
| AIx | without any plaques | 10.50 (9.78–11.21) | Without stenosis | 14.60 (13.95–15.25) | Without IMT thickening | 10.55 (9.83–11.28) | |
| | with plaques | 20.77 (19.81–21.74) | With ICA stenosis >50% | 22.85 (15.15–30.55) | With IMT thickening | 20.35 (19.38–21.31) | |

Aix- augmentation index, PWV–pulse wave velocity, QUICKI -quantitative insulin sensitivity check index, HOMA-IR–homeostatic model assessment, HbA1c – glycated hemoglobin A1c, BMI–body mass index, WHR–waist to hip ratio, LDL-C–low density lipoprotein, ICA–internal carotid artery, IMT–intima media thickening.

might be less related to subclinical atherosclerosis in older adults than in middle-aged adults, which further underlines the importance of our results [24]. In a prior research paper containing data from the Bialystok PLUS cohort, it was shown that FLAIS represents a useful index to assess the cluster of insulin resistance-associated cardiovascular risk factors in general population [25].

According to our findings, HbA1c appears to be a preferable biomarker of early atherosclerotic changes than glucose parameters. Previously, higher blood HbA1c levels were linked to an increased risk of preclinical atherosclerosis in those at low CVR, while there was no association in those at moderate risk [26]. A study using dynamic contrast-enhanced plaque imaging

**Table 3. Regression models performed for the whole study population.** The dependent variable was the presence of atherosclerotic plaques. Well-known cardiovascular risk factors were replaced by cardiovascular risk categories.

| | Full model | | | Reduced model | | |
|---|---|---|---|---|---|---|
| | OR | 95% CI for OR | | p | OR | 95% CI for OR | | p |
| Prediabetes | 1.563 | 1.093 | 2.236 | 0.014 | 1.692 | 1.195 | 2.394 | 0.003 |
| CV risk | | | | <0.001 | | | | <0.001 |
| CV risk 2 vs 1 | 5.903 | 3.917 | 8.896 | <0.001 | 6.033 | 4.009 | 9.077 | <0.001 |
| CV risk 3 vs 1 | 16.408 | 7.941 | 33.903 | <0.001 | 16.968 | 8.193 | 35.143 | <0.001 |
| Brachial-ankle PWV | 1.184 | 1.069 | 1.312 | 0.001 | 1.192 | 1.075 | 1.320 | 0.001 |
| AIx | 1.060 | 1.043 | 1.078 | <0.001 | 1.060 | 1.043 | 1.077 | <0.001 |
| Serum insulin level after 2h | 1.003 | 1.000 | 1.006 | 0.074 | | | | |
| Nagelkerke $R^2$ | 0.51 | | | | 0.50 | | | |
| Model specificity | 88.89% | | | | 87.56% | | | |
| Model sensitivity | 69.34% | | | | 68.87% | | | |

CV–cardiovascular, PWV–pulse wave velocity, Aix- augmentation index, CI–Confidence interval, OR–odds ratio.

showed that leaky plaque neovascularization correlated with HbA1c levels and may be associated with faster progression of atherosclerosis in diabetic patients [27]. Moreover, elevating HbA1c is separately connected with the occurrence of high-risk plaque in non-diabetic individuals even if LDL-C was controlled by statin therapy [28]. Wang et al. showed that HbA1c level was an independent indicator of poor functional outcome in patients with acute anterior circulation ischemic stroke, especially with the large-artery atherosclerosis subtype [29]. The reported data, confirmed by our results, suggest that the inclusion of HbA1c in CVR calculators may prove clinically beneficial.

Our outcomes showed that both, early and more advanced atherosclerotic lesions, occur in non-diabetic hyperglycemic patients more often than in normoglycemic subjects. Moreover, a published literature review outlined that effective glycemic control, especially in earlier stages of DM, reduces the progression of subclinical atherosclerosis, which was assessed by the coronary artery calcification, carotid IMT and arterial stiffness [29, 30]. Other researchers have found that non-invasive early atherosclerotic parameters such as epicardial fat thickness and carotid IMT are increased in patients with prediabetes (especially with IGT) and could be useful indicators of ASCVD risk in this group [31, 32]. We also highlight the use of non-invasive measurements–carotid ultrasound or PWV–as parameters that may be particularly useful in early identification of patients with prediabetes and increased risk of symptomatic ASCVD in the future. Another cross-sectional study showed high carotid plaque presence and burden in new-onset T2DM subjects, especially in women [33] and indicates earlier preventive interventions in this group of patients.

Limited evidence comes from recent research about prevalence of moderate hyperglycemia. Referring to data from the International Diabetes Federation, an estimated 17% of the world's population is diagnosed with non-diabetic hyperglycemia [4], which is less than half of what is presented in our study. Recently, another population-based study also indicated an alarming rise in prediabetes in the population—30.29% in the Hoveyzeh Cohort Study (n = 10,009), especially in overweight and obese patients [34, 35]. Our previously reported research revealed that more than half of the general population has impaired glucose metabolism (including 40% of prediabetes) [18]. An unusually high prevalence of prediabetes (approximately 5%) has been shown in a group of children between 6 and 10 years old, indicating the need for prevention even at a young age [36]. There was also a very high prevalence of prediabetes at 72.3% in patients with chronic coronary syndrome [37]. The moment of diagnosis of prediabetes provides a therapeutic window to incorporate early lifestyle or pharmacological interventions to prevent cardiovascular events.

There is a consensus that the cutoff identifying prediabetes using HbA1c is 5.7% [38, 39]. In our study, we used both OGTT and hemoglobin HbA1c to diagnose prediabetes–using HbA1c alone could underestimate the group with hyperglycemia. Another very large population-based study used only HbA1c to diagnose prediabetes and estimated a prevalence of 6.4%—less than our findings which might be due to the used method and particular, lower risk population [40]. We believe that an OGTT and HbA1c should be used to actively search for prediabetes in the general population. In addition, elevated HbA1c values may be an indicator of higher ASCVD risk in cardiovascular risk category 1 and may prompt further non-invasive diagnostics for subclinical atherosclerosis.

In the study, a single fasting glucose measurement was performed and therefore our population of patients with diabetes may be underestimated. In addition, some patients did not have the OGTT performed for technical reasons–inability to draw blood or vomiting reflex when consuming dissolved glucose. We used GFR to classify patients for cardiovascular risk. This is a cross-sectional analysis of the data and therefore causality cannot be inferred. Despite these limitations of our study, we believe that both the large sample and the thorough data

analysis performed indicate the value of the publication. Our findings may have implications for the management of patients with preclinical conditions such as prediabetes and asymptomatic carotid atherosclerosis. Our study suggests that expansion of outpatient care in higher-risk patients who are not included in the risk calculators proposed in the recommendations. SCORE-2 does not include prediabetes and thus may underestimate the risk of preclinical atherosclerosis. Our results support this conclusion, as prediabetes significantly contributes to the risk of atherosclerosis when analyzed with CVR classes and provides additional pivotal information. Once prediabetes, especially IGT, is diagnosed, the GP should suggest lifestyle changes and increased physical activity and intensify follow-up visits to observe the development of preclinical atherosclerosis.

## Conclusions

The study demonstrates the high prevalence of non-diabetic hyperglycemia in the general population, which is associated with the occurrence of preclinical atherosclerosis, especially in the low to moderate cardiovascular risk category. The more advanced preclinical atherosclerotic lesions are also significantly more common in the prediabetic population. Our study suggests that cardiovascular risk may be underestimated in people with prediabetes. The identification of this group in an apparently low-risk population may improve appropriate risk assessment and help implement early interventions to prevent cardiovascular incidents. Further studies on predictors of subclinical atherosclerosis in prediabetes are needed.

## Supporting information

**S1 Checklist. Human participants research checklist.**
(DOCX)

**S1 Fig. Receiver operating characteristic (ROC) curves; larger values of the test result variable indicate stronger evidence for a positive actual state.** Dependent variable: the presence of any atherosclerotic plaques on ultrasound of the carotid arteries.
(TIFF)

**S1 Table. Receiver operating characteristic.**
(TIFF)

**S2 Table. Prevalence of prediabetes in the general population with cardiovascular risk categories and the presence of atherosclerotic plaques in the carotid arteries.** P-values were derived from Chi-square tests.
(TIFF)

**S3 Table. Prevalence of prediabetes in the general population with cardiovascular risk categories and the presence of atherosclerotic plaques in the carotid arteries (included subpopulation IFG+IGT alone).** P-values were derived from Chi-square tests.
(TIFF)

**S4 Table. The occurrence of a stenosis of more than 50% in the right or left internal carotid artery in the study population considering glucose metabolism disturbances.**
(TIFF)

**S5 Table. Assessment of preclinical atherosclerotic progression based on the level of stenosis of the right or left internal carotid artery (ICA).** P-values were derived from Chi-square tests.
(TIFF)

## Author Contributions

**Conceptualization:** Natalia Anna Zieleniewska, Irina Kowalska, Karol Adam Kamiński.

**Formal analysis:** Natalia Anna Zieleniewska, Jacek Jamiołkowski.

**Funding acquisition:** Natalia Anna Zieleniewska.

**Investigation:** Natalia Anna Zieleniewska, Małgorzata Chlabicz, Adam Łukasiewicz, Marlena Dubatówka, Marcin Kondraciuk, Paweł Sowa.

**Methodology:** Natalia Anna Zieleniewska, Jacek Jamiołkowski, Irina Kowalska, Karol Adam Kamiński.

**Resources:** Natalia Anna Zieleniewska, Karol Adam Kamiński.

**Supervision:** Karol Adam Kamiński.

**Validation:** Karol Adam Kamiński.

**Visualization:** Natalia Anna Zieleniewska.

**Writing – original draft:** Natalia Anna Zieleniewska, Karol Adam Kamiński.

**Writing – review & editing:** Małgorzata Chlabicz, Irina Kowalska, Karol Adam Kamiński.

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
