## [Editor Report · Decision Letter 0]

16 Feb 2024

PONE-D-23-42492The impact of prediabetes on preclinical atherosclerosis in general apparently healthy population.PLOS ONE

Dear Dr. Kaminski,

Thank you for submitting your manuscript to PLOS ONE. After careful consideration, we feel that it has merit but does not fully meet PLOS ONE’s publication criteria as it currently stands. Therefore, we invite you to submit a revised version of the manuscript that addresses the points raised during the review process.

*Comments from PLOS Editorial Office:*

*We apologize for the difficulties you were having with your submission. For your convenience, we have summarized the points previously raised by the academic editor below:*

*- Accept all changes and cancel tracking mode*

- Use a uniform font and font size throughout the manuscript including references, tables and figure legends

- Mention all tables (including the supplementary ones) in the main text and sort them according to their order of appearance in the text

- Move the study limitations into the Discussion

- Put the Declarations part on a separate page and arrange it more clearly

- Rearrange the figures and tables so that they are readable: In particular, the authors should put each item on a separate page and format it in a way so that it does not exceed this page.

- Use no colours in the tables

*- Use "." as decimal separators in all tables*

We look forward to receiving your revised manuscript.

Kind regards,

Andreas Beyerlein

Academic Editor

PLOS ONE

Additional Editor Comments:

Please revise the files as discussed.

---

## [Author Response · Author response to Decision Letter 0]

19 Feb 2024

17th February 2024

Dear Editor

We would like to thank you for your helpful comments on the manuscript The impact of prediabetes on preclinical atherosclerosis in general apparently healthy population. We have edited the manuscript to address all the concerns. All the changes we have made are marked in red.

We hope that after corrections suggested by the Editorial Office, the manuscript is now suitable for publication in the PLOS ONE.

We are looking forward to hearing from you soon.

Yours sincerely

Karol Kamiński

On behalf of all authors.

Comments from PLOS Editorial Office:

We apologize for the difficulties you were having with your submission. For your convenience, we have summarized the points previously raised by the academic editor below:

- Accept all changes and cancel tracking mode.

The changes have been accepted. Tracking mode has been canceled.

- Use a uniform font and font size throughout the manuscript including references, tables and figure legends.

The font has been standardized to Times New Roman, font size 12.

- Mention all tables (including the supplementary ones) in the main text and sort them according to their order of appearance in the text

Tables and figures are mentioned in the main text of the manuscript.

- Move the study limitations into the Discussion

Study limitations have been moved to the last paragraph of the Discussion.

- Put the Declarations part on a separate page and arrange it more clearly

The declarations were placed on page 2 of the manuscript and prepared in accordance with the journal's recommendations.

- Rearrange the figures and tables so that they are readable: In particular, the authors should put each item on a separate page and format it in a way so that it does not exceed this page.

The figures were prepared in accordance with these rules. The tables have been reformatted according to this recommendation, but Tables 1 and 3 contains too much information to fit on one page.

- Use no colours in the tables

Well, it has been corrected.

- Use "." as decimal separators in all tables

Commas have been changed to dots.

---

## [Decision Letter · Decision Letter 1]

8 May 2024

PONE-D-23-42492R1The impact of prediabetes on preclinical atherosclerosis in general apparently healthy population.PLOS ONE

Dear Dr. Kaminski,

Thank you for submitting your manuscript to PLOS ONE. After careful consideration, we feel that it has merit but does not fully meet PLOS ONE’s publication criteria as it currently stands. Therefore, we invite you to submit a revised version of the manuscript that addresses the points raised during the review process.

We look forward to receiving your revised manuscript.

Kind regards,

Andreas Beyerlein

Academic Editor

PLOS ONE

**Additional Editor Comments:**

General:

- The authors should make sure that their manuscript follows the STROBE guidelines.

- Both the title and the abstract sholud mention the cross-sectional study design.

- At some occasions, the results are presented in present tense. Instead, all results should be described in past tense.

- The authors present a bunch of similar analyses on the same dataset, leading to issues of multiple testing. Preferably, only those results which are directly related to the main hypothesis should be shown, while several side analyses might just be left out. I added some suggestions below. Additionally, p-values and interpretations of statistical significance should be presented only for the most relevant associations.

- In the spirit of Open and Reproducible Science, the analysis code should be made available in an online repository together with a data dictionary, and the respective URL should be mentioned in the Methods section.

Abstract:

- The abstract should mention setting, locations, periods of recruitment, duration of follow-up, and age of the participants.

- The sentence "In multivariate logistic regression, prediabetes significantly increased risk for preclinical atherosclerosis (p=0.015)." should be rephrased to "...was associated with significantly increased risk...", and OR with 95% CI should be shown instead of a p-value.

- It seems not correct to state that the analysis included 1866 participants given that a large proportion of them were excluded due to technical reasons or prevalent DM, and it also does not fit to the statement ..."more than half of the study participants (n=797, 51.9%)"

Introduction:

- "...and 319 million adults (6.2%) with IFG" should be rephrased as "... to have IFG".

- When introduced, abbreviations should be used throughout the manuscript, which is e.g. not the case in the sentence "First, the presence of nondiabetic hyperglycemia signifies an increased risk of developing type 2 diabetes..."

- The study hypothesis is not properly motivated. For example, the authors state that prediabetes is associated with a higher risk of cardiovascular disease. So would it not just be expected that it is also associated with a higher risk of preclinical atherosclerosis? Are there already any other studies on this topic?

Methods:

- In the sentence "We randomly sampled citizens in such numbers as to obtain a distribution of proportions in terms of age and gender similar to that of the city's population.", what is meant by "in such numbers"? Please add that this sentence only pertains to the age range mentioned before.

- What is meant by "research tests"?

- In the Matsuda formula, a bracket seems to be missing. Can the abbreviations I0 and G0 also be used in the HOMA-IR and Matsuda formulas?

- "... for checking variable normality." should be revised to "... for checking whether the variables followed a normal distribution."

- Supplementary Tables 1 and 2 seem superfluous, as these definitions can easily be mentioned in the main text in the methods section.

Results:

- Please add a descriptive table 1 to characterize the whole study population (excluding the participants excluded).

- Supplementary Figure 2 seems superfluous, the more so as it is not mentioned in the main text.

- The current tables 1 and 3 are heavily overloaded and need to be condensed or split into two separate tables each (e.g. with and without prediabetes markers). The p-values should be removed, as they are sensitive to sample size and prone to multiple testing. Instead, I would suggest to show mean and 95% confidence intervals instead of median and IQR (see also the comments of Reviewer 1 concerning this).

- The sentence "We used the diagnostic criteria for DM and prediabetes..." seems out of place in the results section.

- The numbers in Suppl. Figure 1 do not add up to 1866. The authors should show a proper flowchart as requested in the STROBE guidelines (which would not include the numbers of participants with or without prediabetes).

- Did the persons excluded due to technical reasons differ from the remaining population with respect to age, sex or other characteristics? If so, potential bias by exclusion of these participants needs to be discussed.

- "The population of attendees without dysglycemia consisted of 634 patients (41.2%). Importantly, the population with prediabetes accounted for more than half of the study participants (n=797, 51.85%)." Why do these two numbers not add up to 100%? As a minor point, percentages should presented with the same number of digits throughout the whole manuscript (main text and tables).

- "Importantly, the population without glucose metabolism disturbances": Leave out "Importantly" (there should be no interpretation in the results section)

- What is meant by "a statistically relevant difference"?

- All supplementary tables should be named by the order of their appearance in the main text.

- The abbreviation USG should be explained in the main text and each figure / table where it appears.

- The legends of Figure 1 and Supplementary Table 5 should not be selective in mentioning the AUC values of the predictors, i.e. show all or none of them.

- "most statistically significant indicators" is wrong wording. A result can either be statistically significant or not.

- Why does Supplementary Table 5 not include glucose level after 1 h, and why is the AUC of HbA1c different in the table to the AUC mentioned in the main text?

- Table 2 is difficult to read and might rather be replaced by e.g. stacked barplots.

- It is confusing that some tables are arranged with CVR as columns and glucose metabolism and rows and vice versa. The appearance of these tables should be unified.

- Supplementary Tables 4 and 12: The p** values should be removed.

- The logistic regression analyses are confusing and contradictory in their results, which is particularly irritating given that there seem to be quite strong associations between prediabetes and preclinical atherosclerosis in bivariate analyses. The results from Supplementary Table 8 seem to be based on the most reasonable approach (while the other regression analyses seem superfluous), but they indicate a ngeative association between the two factors. Can the authors please comment on this issue?

Discussion:

- The discussion may need to be revised in accordance with the revision of the logistic regression analyses.

- The authors applied a large number of tests, but did not correct for multiple testing. This should be clearly stated in the discussion together with the consequence that their results should be seen as hypothesis-generating and need confirmation from other studies.

- "This study confirms a very high prevalence of prediabetes in a general population." The statements in this paragraph and also in the conclusion are too general and need to be more specific to the underlying population. Further, the authors might discuss how generalizable their results are with respect to other populations in Europe and across the world.

Reviewers' comments:

Reviewer's Responses to Questions

**Comments to the Author**

1. If the authors have adequately addressed your comments raised in a previous round of review and you feel that this manuscript is now acceptable for publication, you may indicate that here to bypass the “Comments to the Author” section, enter your conflict of interest statement in the “Confidential to Editor” section, and submit your "Accept" recommendation.

Reviewer #1: (No Response)

Reviewer #2: (No Response)

Reviewer #3: (No Response)

2. Is the manuscript technically sound, and do the data support the conclusions?

Reviewer #1: Yes

Reviewer #2: Yes

Reviewer #3: Yes

3. Has the statistical analysis been performed appropriately and rigorously? 

Reviewer #1: No

Reviewer #2: I Don't Know

Reviewer #3: Yes

4. Have the authors made all data underlying the findings in their manuscript fully available?

Reviewer #1: No

Reviewer #2: Yes

Reviewer #3: Yes

5. Is the manuscript presented in an intelligible fashion and written in standard English?

Reviewer #1: Yes

Reviewer #2: Yes

Reviewer #3: Yes

6. Review Comments to the Author

Reviewer #1: This manucript examines the impact of prediabetes on preclinical atherosclerosis in general apparently healthy

population in Poland.

The paper is well written and organised.

Please find my comments below:

- Statistical analysis section: It is stated that continuous variables are described using medians (IQR). However, in Table 1, it seems that some variables (i.e. Age) are described using mean +/- sd. This should be clarified in the statistical analysis section as well as the results for this.

- Statistical analysis section: Shapiro-Wilks test is sensitive to large sample sizes and the null hypothesis can be rejected even with small deviations from normality. Kolmogorov-Smirnov is a better option when sample size is >50. However, here the sample size is quite large so the central limit theorem holds. The authors could have proceeded with using only parametric tests.

- Fig.2: SPSS 27 does not produce forest plots. The authors should state clearly the software used for the data analysis and visualisation.

- Any p-values=0.000 should be changed to <0.001.

Reviewer #2: The manuscript entitled "The impact of prediabetes on preclinical atherosclerosis in an apparently healthy general population" addresses a highly interesting and significant clinical issue. Early identification of individuals with dysglycemia is essential for the effective implementation of cardiovascular and metabolic prevention measures. Considering the authors' understanding of the research problem, the reviewer requests a discussion on the practical application of the obtained results - whether this data is sufficient for updating/altering recommendations, particularly for General Practitioners, and what those changes should involve?

Reviewer #3: The paper PONE-D-23-42492R1 » The impact of prediabetes on preclinical atherosclerosis in general apparently healthy population« describes the relation of prediabetes (diagnosed by fasting glucose levels or by oral glucose tolerance test) and the presence of carotid artery plaques in a random sample of 1866 subjects from the Bialystok PLUS study. Prediabetes significantly increased risk for preclinical carotid atherosclerosis according to a multivariate logistic regression model.

I have the following comments:

Materials and methods, Assessment of insulin sensitivity. Please, list thel criteria for diagnosing prediabetes.

Typographical corrections:

Abstract, line 9: Atherosclerotic plaques in the carotid arteries were …

Fig. 1. The x-axis should be labeled »1-specificity«.

7. PLOS authors have the option to publish the peer review history of their article (what does this mean?). If published, this will include your full peer review and any attached files.

Reviewer #1: No

Reviewer #2: No

Reviewer #3: No

---

## [Author Response · Author response to Decision Letter 1]

18 Jun 2024

General:

- The authors should make sure that their manuscript follows the STROBE guidelines.

Thank you for suggestion. We have corrected the manuscript accordingly and clearly stated this in the manuscript. We have created a revised Flowchart (Figure 1 Supplementary Material). We have described the population study method in detail. Moreover, our data has been recalculated and completed, due to numerous statistical comments. 

- Both the title and the abstract should mention the cross-sectional study design.

I added in the title: The impact of prediabetes on preclinical atherosclerosis in general apparently healthy population: a cross-sectional study.

I added also in Abstract, Methods: The cross-sectional analysis included 1431 participants of a population-based cohort study. Comprehensive biochemical assessments were performed.

- At some occasions, the results are presented in present tense. Instead, all results should be described in past tense.

I have changed the tense as suggested.

- The authors present a bunch of similar analyses on the same dataset, leading to issues of multiple testing. Preferably, only those results which are directly related to the main hypothesis should be shown, while several side analyses might just be left out. I added some suggestions below. Additionally, p-values and interpretations of statistical significance should be presented only for the most relevant associations.

Thank you for your comment. We have made the corrections as suggested below.

- In the spirit of Open and Reproducible Science, the analysis code should be made available in an online repository together with a data dictionary, and the respective URL should be mentioned in the Methods section.

We did not used pre-programmed scripts. We have done ad hoc analysis in the SPSS.

Abstract:

- The abstract should mention setting, locations, periods of recruitment, duration of follow-up, and age of the participants.

The following sentence has been added:

The analysis was a part of the Bialystok PLUS cohort study and represented a random sample of Bialystok residents aged 20-79 at the time of sampling (July 2017-January 2023). 

- The sentence "In multivariate logistic regression, prediabetes significantly increased risk for preclinical atherosclerosis (p=0.015)." should be rephrased to "...was associated with significantly increased risk...", and OR with 95% CI should be shown instead of a p-value.

Thank you, it has been corrected.

In multivariate logistic regression, prediabetes was independently associated with significantly increased risk of preclinical atherosclerosis (OR= 1.56, 95% CI 1.09-2.24), along with CVR categories, pulse wave velocity and central blood pressure augmentation index.

- It seems not correct to state that the analysis included 1866 participants given that a large proportion of them were excluded due to technical reasons or prevalent DM, and it also does not fit to the statement ..."more than half of the study participants (n=797, 51.9%)"

In accordance with the Editor's comments, we have included the correct flowchart of study population (Figure 1 Supplementary Materials) and changed the number given as size of the population to 1431. 

The cross-sectional analysis included 1431 participants of a population-based study (mean age 48.82).

Introduction:

- "...and 319 million adults (6.2%) with IFG" should be rephrased as "... to have IFG"

- When introduced, abbreviations should be used throughout the manuscript, which is e.g. not the case in the sentence "First, the presence of nondiabetic hyperglycemia signifies an increased risk of developing type 2 diabetes..."

Both corrections have been applied.

- The study hypothesis is not properly motivated. For example, the authors state that prediabetes is associated with a higher risk of cardiovascular disease. So would it not just be expected that it is also associated with a higher risk of preclinical atherosclerosis? Are there already any other studies on this topic?

Indeed, the introduction was too brief. An additional paragraph has been added to further motivate the research hypothesis. It should be emphasized that there are quite strong controversies about the effect of prediabetes on atherosclerosis and hence cardiovascular risk. Currently European Society of Cardiology guidelines do not incorporate prediabetes in their cardiovascular risk calculators, mainly due to limited evidence from general population. Therefore the evidence our study provides contributes to this scientific discussion.

A previously conducted study of patients with first-time acute coronary syndrome (ACS) who underwent urgent coronary angiography showed that coronary atherosclerosis is more advanced in patients with prediabetes than in patients without DM [38]. Furthermore, the association of the occurrence of coronary atherosclerosis was shown to be comparable between patients with prediabetes and patients with DM at the time of first ACS [38]. Another study using percutaneous coronary intervention showed that coronary lesions in prediabetic patients were associated with higher levels of lipid-rich atherosclerotic plaques [39]. Prediabetes is associated with inflammation and vasoconstriction, which promote atherosclerosis in the coronary arteries [40]. Notably, limited evidence comes from research on both, prediabetes and carotid atherosclerosis in general population.

Methods:

- In the sentence "We randomly sampled citizens in such numbers as to obtain a distribution of proportions in terms of age and gender similar to that of the city's population.", what is meant by "in such numbers"? Please add that this sentence only pertains to the age range mentioned before.

The text has been transformed for better understanding.

Our analysis was a part of the Bialystok PLUS cohort study and represented a random sample of Bialystok residents aged 20-79 at the time of sampling (July 2017-January 2023). Bialystok is a medium-sized city located in eastern Poland with a population of 293,400. The recruitment of participants for the population-based study uses a pseudonymized list of residents of Bialystok obtained from the Local Municipal Office. Annually, we randomly sampled citizens to obtain a distribution of proportions in terms of age and gender reflecting that of the city's population. A more detailed study design was described in previously published paper [11].

- What is meant by "research tests"?

We meant all the tests performed in the course of the study, as they were performed solely for the purpose of the study. For greater clarity, we have changed the research test to “the clinical and biochemical measurements”.

All the clinical and biochemical measurements were conducted by qualified medical personnel.

- In the Matsuda formula, a bracket seems to be missing. Can the abbreviations I0 and G0 also be used in the HOMA-IR and Matsuda formulas?

Thank you for helpful suggestions. Corrections have been made in accordance with it.

Quantitative insulin sensitivity check index (QUICKI) was defined as 1/[log(I0) + log(G0)], where I0 is the fasting insulin (μU/ml), and G0 is the fasting glucose (mmol/l) [16]. Homeostasis model assessment of insulin resistance (HOMA-IR) was calculated using the formula: I0  ×  G0 /22.5 [15]. When assessing insulin sensitivity, we used the Matsuda index calculated from the formula 10000/ √ (G0 mg/dl x I0 μU/ml) x (Mean OGTT glucose concentration mg/dl × mean OGTT insulin concentration mg/dl) [17].

- "... for checking variable normality." should be revised to "... for checking whether the variables followed a normal distribution."

Done. 

- Supplementary Tables 1 and 2 seem superfluous, as these definitions can easily be mentioned in the main text in the methods section.

The tables have been removed. A description has been added in Methods, Assessment of Insulin sensitivity.

Results:

- Please add a descriptive table 1 to characterize the whole study population (excluding the participants excluded).

A description of the whole study population has been added to Table 1, in the last column ‘Total’.

- Supplementary Figure 2 seems superfluous, the more so as it is not mentioned in the main text.

Indeed, according to this Supplementary Figure 2 has been deleted. 

- The current tables 1 and 3 are heavily overloaded and need to be condensed or split into two separate tables each (e.g. with and without prediabetes markers). The p-values should be removed, as they are sensitive to sample size and prone to multiple testing. Instead, I would suggest to show mean and 95% confidence intervals instead of median and IQR (see also the comments of Reviewer 1 concerning this).

In accordance with this comment and the suggestions of Reviewer 1, all tables have been revised.

- The sentence "We used the diagnostic criteria for DM and prediabetes..." seems out of place in the results section.

It has been moved to Methods section.

- The numbers in Suppl. Figure 1 do not add up to 1866. The authors should show a proper flowchart as requested in the STROBE guidelines (which would not include the numbers of participants with or without prediabetes).

Figure 1 Supplementary Material has been modified.

- Did the persons excluded due to technical reasons differ from the remaining population with respect to age, sex or other characteristics? If so, potential bias by exclusion of these participants needs to be discussed.

Variable Excluded population for technical reasons n=183

Mean, 95% CI Study population

n=1431

Mean, 95% CI Total

n=1866

Mean, 95% CI

Age 49.67 (47.49-51.85) 46.82 (46.05-47.59) 47.14 (46.41-47.87)

BMI 27.29 (26.24-28.33) 26.30 (26.06-26.54) 26.39 (26.15-26.62)

WHR 0.85 (0.84-0.87) 0.86 (0.85-0.86) 0.86 (0.85-0.86)

SBP 118.08 (115.10-121.06) 122.18 (121.31-123.04) 121.85 (121.02-122.68)

DBP 79.56 (77.73-81.39) 80.48 (79.96-80.99) 80.40 (79.91-80.90)

Total cholesterol 196.56 (188.78-204.33) 193.63 (191.59-195.67) 193.86 (191.89-195.84)

LDL-C 124.30 (117.27-131.32) 123.86 (122.00-125.72) 123.89 (122.09-125.69)

Fasting glucose 97.38 (95.55-99.21) 97.87 (97.38-98.36) 97.83 (97.35-98.30)

HbA1c 5.42 (5.35-5.49) 5.36 (5.34-5.38) 5.36 (5.34-5.38)

Sex, %male 29.0% (22.9%-35.9%) 44.9% (42.4%-47.5%) 43.1% (40.7%-45.6%)

The presence of atherosclerotic plaque 43.0% (34.7%-51.6%) 40.7% (38.2%-43.2%) 40.9% (38.4%-43.3%)

Cholesterol-lowering treatment 75.6% (65.8%-83.3%) 78.7% (75.4%-81.7%) 78.3% (75.2%-81.2%)

In the table we have included a comparison between those excluded for technical reasons (n=183) and the analysed population (n=1431). The values of the parameters we were able to compare are not considerably different from each other. We believe that, despite the statistical difference, the compared populations do not differ clinically.

- "The population of attendees without dysglycemia consisted of 634 patients (41.2%). Importantly, the population with prediabetes accounted for more than half of the study participants (n=797, 51.85%)." Why do these two numbers not add up to 100%? As a minor point, percentages should presented with the same number of digits throughout the whole manuscript (main text and tables).

Percentages have been rounded to 1 decimal places in the texts. Previously, the percentages were counted to the whole group (including those who were excluded). For a better understanding of the analysis, changes have been made.

In further analysis, we considered 1431 participants. The population of attendees without dysglycemia consisted of 634 patients (44.3%). The population with prediabetes accounted for more than half of the study participants (n=797, 55.7%).

- "Importantly, the population without glucose metabolism disturbances": Leave out "Importantly" (there should be no interpretation in the results section)

Thank you.

- What is meant by "a statistically relevant difference"?

We have changed to “statistically significant difference”.

- All supplementary tables should be named by the order of their appearance in the main text.

Done.

- The abbreviation USG should be explained in the main text and each figure / table where it appears.

Done. 

- The legends of Figure 1 and Supplementary Table 5 should not be selective in mentioning the AUC values of the predictors, i.e. show all or none of them.

OK, the AUC values have been removed from the legends.

- "most statistically significant indicators" is wrong wording. A result can either be statistically significant or not.

Indeed. This has been corrected.

We observed that all examined parameters proved to be statistically significant indicators of the presence of atherosclerotic plaques in carotid ultrasound, also upon application of Bonferroni correction. However, the highest AUC values were obtained for serum glucose concentration after 1h and HbA1c (AUC=0.73; 95% CI 0.70-0.76 for glucose level after 1 h and AUC=0.72; 95% CI 0.69-0.75 for HbA1c). 

- Why does Supplementary Table 5 not include glucose level after 1 h, and why is the AUC of HbA1c different in the table to the AUC mentioned in the main text?

Many thanks for your thorough review of the manuscript. Table 5 Supplementary Materials has been corrected. The values entered in the Manuscript were correct. This was a mistake in transferring data from a statistical programme to a word document.

- Table 2 is difficult to read and might rather be replaced by e.g. stacked barplots.

Table 2 has been transformed into stacked barplots. 

- It is confusing that some tables are arranged with CVR as columns and glucose metabolism and rows and vice versa. The appearance of these tables should be unified.

We wanted to emphasise what exactly is being compared in the tables by using this approach. These are additional analyses and have therefore been placed in Supplementary Materials.

- Supplementary Tables 4 and 12: The p** values should be removed.

Done.

- The logistic regression analyses are confusing and contradictory in their results, which is particularly irritating given that there seem to be quite strong associations between prediabetes and preclinical atherosclerosis in bivariate analyses. The results from Supplementary Table 8 seem to be based on the most reasonable approach (while the other regression analyses seem superfluous), but they indicate a negative association between the two factors. Can the authors please comment on this issue?

To sum up, bivariate analyses suggest that there are positive associations between prediabetes and preclinical atherosclerosis. In the multivariate analyses, we used two separate approaches. In the first multivariate model, we included individual classical CVR factors (age, sex, SBP, total cholesterol, LDL-C, cigarette smoking, BMI) as co-variates. In this model, we did not show a statistically significant association between prediabetes and the occurrence of atherosclerotic plaques, which may be due to strong association of prediabetes with other risk factors (age, blood pressure, BMI) and the fact that they may have a strong association with preclinical plaques. In the second approach, we used a cardiovascular risk scale considering the cardiovascular risk factors in a synthetic way proposed by the European Society of Cardiology. In this model, we present an independent positive relationship (Table 10, Supplementary Materials) between prediabetes and the presence of preclinical carotid artery lesions (95% CI 1.091-2.232, OR 1.561), which emphasizes potential value of including prediabetes in cardiovascular risk assessment. Model 2 informs us that the prediabetes in addition to CVR categories is associated with a higher risk of preclinical atherosclerosis, which may be relevant for physicians in clinical practice. 

Discussion:

- The authors applied a large number of tests, but did not correct for multiple testing. This should be clearly stated in the discussion together with the consequence that their results should be seen as hypothesis-generating and need confirmation from other studies.

Thank you for this comment. In our case, we tested the hypothesis whether prediabetes increases the risk of preclinical atherosclerosis. Additional analyses were hypothesis generating and assessed the mechanism of this association. A Bonferroni cor

---

## [Decision Letter · Decision Letter 2]

10 Jul 2024

PONE-D-23-42492R2The impact of prediabetes on preclinical atherosclerosis in general apparently healthy population: a cross-sectional studyPLOS ONE

Dear Dr. Kaminski,

Thank you for submitting your manuscript to PLOS ONE. After careful consideration, we feel that it has merit but does not fully meet PLOS ONE’s publication criteria as it currently stands. Therefore, we invite you to submit a revised version of the manuscript that addresses the points raised during the review process.

We look forward to receiving your revised manuscript.

Kind regards,

Andreas Beyerlein

Academic Editor

PLOS ONE

Journal Requirements:

Additional Editor Comments:

The authors have improved their manuscript considerably. Only a few points remain to be addressed:

General:

- There are still some results described in present tense, e.g. those under "Atherosclerotic plaque presence in particular cardiovascular risk categories". The whole manuscript should be checked and revised again regarding this issue.

- Throughout the whole manuscript (including tables and figures), only black font should be used.

- Please re-check and confirm that all numbers mentioned in the main text fit to the numbers presented in the tables and figures.

Abstract:

- Please add "years" after 20-79 and 48.82 (also in the main text), and mention that Bialystok is a city from Poland.

- Please omit "the" in "proved to be the statistically significant indicators".

Results:

- Figure 1 should be moved from the main document to the Supplement, and vice versa for Supplementary Figure 1 (flowchart).

- Suggest to show vertical bars in Figure 2 and to remove the yellow bars (sorry, my previous comment hinting to a "stacked barplot" was somewhat misleading).

- Suggest to move the "Total column" in Table 1 to the very left.

- Supplementary Table 1 seems superfluous, as these numbers might just be described in the main text.

- "Additionally, 166 patients were excluded from further analysis for technical reasons...". This number cannot be found in the flowchart. Please explain.

- Suggest to revise "In further analysis we considered 1431 participants." by e.g. "After applying these exclusion criteria, the data of 1431 participants remained for analysis."

- It makes no sense to present the Supplementary Tables 5 and 6 as they are likely to be prone to collinearity issues as mentioned in the text. Instead, the choice (and exclusion) of the predictors in Supplementary Table 7 should be properly motivated and described in detail, possibly accompanied by a stepwise variable selection process. The predictor prediabetes should be shown at the top of this table. Why were sex and age not included here?

- Finally, it would seem more appropriate to move Supplementary Table 7 to the main results and to remove Figure 3 instead.

Reviewers' comments:

Reviewer's Responses to Questions

**Comments to the Author**

1. If the authors have adequately addressed your comments raised in a previous round of review and you feel that this manuscript is now acceptable for publication, you may indicate that here to bypass the “Comments to the Author” section, enter your conflict of interest statement in the “Confidential to Editor” section, and submit your "Accept" recommendation.

Reviewer #1: All comments have been addressed

Reviewer #2: All comments have been addressed

Reviewer #3: All comments have been addressed

2. Is the manuscript technically sound, and do the data support the conclusions?

Reviewer #1: Yes

Reviewer #2: Yes

Reviewer #3: Yes

3. Has the statistical analysis been performed appropriately and rigorously? 

Reviewer #1: Yes

Reviewer #2: I Don't Know

Reviewer #3: Yes

4. Have the authors made all data underlying the findings in their manuscript fully available?

Reviewer #1: Yes

Reviewer #2: Yes

Reviewer #3: Yes

5. Is the manuscript presented in an intelligible fashion and written in standard English?

Reviewer #1: (No Response)

Reviewer #2: Yes

Reviewer #3: Yes

6. Review Comments to the Author

Reviewer #1: The authors have replied to my comments adequately. I only have one minor comment. Please indicate the dependent variable in the captions of Tables 5 and 6.

Reviewer #2: (No Response)

Reviewer #3: The manuscript is now substantially improved. The reviewers' comments have been adequately addressed.

7. PLOS authors have the option to publish the peer review history of their article (what does this mean?). If published, this will include your full peer review and any attached files.

Reviewer #1: No

Reviewer #2: No

Reviewer #3: No

---

## [Author Response · Author response to Decision Letter 2]

16 Jul 2024

Dear Editor 

We would like to thank you for your comments on the manuscript The impact of prediabetes on preclinical atherosclerosis in general apparently healthy population: a cross-sectional study. We have edited the manuscript to address all the concerns. All the changes we have made are marked in red.

We hope that after corrections suggested by the reviewer, the manuscript is now suitable for publication in the PLOS One Journal.

We are looking forward to hearing from you soon.

On behalf of all authors.

Yours sincerely

Karol Kamiński

Journal Requirements:

The list has been checked. All references are correct and have not been changed or retracted.

Additional Editor Comments:

The authors have improved their manuscript considerably. Only a few points remain to be addressed:

General:

- There are still some results described in present tense, e.g. those under "Atherosclerotic plaque presence in particular cardiovascular risk categories". The whole manuscript should be checked and revised again regarding this issue.

Indeed, the tense used has been corrected.

- Throughout the whole manuscript (including tables and figures), only black font should be used.

Done.

- Please re-check and confirm that all numbers mentioned in the main text fit to the numbers presented in the tables and figures.

Done.

Abstract:

- Please add "years" after 20-79 and 48.82 (also in the main text), and mention that Bialystok is a city from Poland.

Done. 

- Please omit "the" in "proved to be the statistically significant indicators".

Thank you, corrected. 

Results:

- Figure 1 should be moved from the main document to the Supplement, and vice versa for Supplementary Figure 1 (flowchart).

Done. 

- Suggest to show vertical bars in Figure 2 and to remove the yellow bars (sorry, my previous comment hinting to a "stacked barplot" was somewhat misleading).

Done. 

- Suggest to move the "Total column" in Table 1 to the very left.

Done. 

- "Additionally, 166 patients were excluded from further analysis for technical reasons...". This number cannot be found in the flowchart. Please explain.

Thank you, it was a typing mistake. Everything now corresponds with Flowchart.

The study enrolled 1866 participants. According to STROBE guidelines, the flowchart represents the study population (Figure 1 Supplementary Materials). Prevalent diabetes (n=252, 13.5%) was diagnosed on the basis of medical history (n=140) and glucose levels after 2h OGGT (n=112). All prevalent diabetic patients were excluded from further analysis. Additionally, 183 patients were excluded from further analysis for technical reasons – inability to draw blood, perform an OGTT or US of the carotid arteries [31]. After applying these exclusion criteria, the data of 1431 participants remained for analysis.

- Suggest to revise "In further analysis we considered 1431 participants." by e.g. "After applying these exclusion criteria, the data of 1431 participants remained for analysis."

Ok.

- It makes no sense to present the Supplementary Tables 5 and 6 as they are likely to be prone to collinearity issues as mentioned in the text. Instead, the choice (and exclusion) of the predictors in Supplementary Table 7 should be properly motivated and described in detail, possibly accompanied by a stepwise variable selection process. The predictor prediabetes should be shown at the top of this table. Why were sex and age not included here?

That's fine, in that case we deleted Table 5 and Table 6 Supplementary Materials. Gender and age were not included because they are already included in the cardiovascular risk categories. We used SCORE calculators to calculate CVR categories, which take gender and age into account. We decided to exclude these variables to avoid potential multicollinearity problem in regression model. We have changed the description in the main text.

We also performed multivariate logistic regression model where the dependent variable was the presence of any atherosclerotic plaques (Table 3). We built a model in which we included CVR categories (which already include classic cardiovascular risk factors, including gender and age), glucose metabolism and vascular stiffness parameters. Figure 2 Supplementary Materials is a graphical summary of the fully adjusted logistic regression model. Prediabetes was associated with significantly increased risk of preclinical atherosclerosis (OR= 1.56, 95% CI 1.09-2.24; p=0.014), along with CVR categories, pulse wave velocity and central blood pressure augmentation index. 

- Finally, it would seem more appropriate to move Supplementary Table 7 to the main results and to remove Figure 3 instead.

Done.

Reviewer #1: The authors have replied to my comments adequately. I only have one minor comment. Please indicate the dependent variable in the captions of Tables 5 and 6.

Thank you. According to the editor's comments, these tables have been deleted from the supplement. We added caption in Table 3 of the main text considering the dependent variable.

Reviewer #3: The manuscript is now substantially improved. The reviewers' comments have been adequately addressed.

Thank you.

---

## [Editor Report · Decision Letter 3]

19 Jul 2024

PONE-D-23-42492R3The impact of prediabetes on preclinical atherosclerosis in general apparently healthy population: a cross-sectional studyPLOS ONE

Dear Dr. Kaminski,

Thank you for submitting your manuscript to PLOS ONE. After careful consideration, we feel that it has merit but does not fully meet PLOS ONE’s publication criteria as it currently stands. Therefore, we invite you to submit a revised version of the manuscript that addresses the points raised during the review process.

We look forward to receiving your revised manuscript.

Kind regards,

Andreas Beyerlein

Academic Editor

PLOS ONE

Journal Requirements:

Additional Editor Comments:

The manuscript is now almost ready for journal acceptance. Please accept all changes made so far so that only changes required for this (hopefully last) revision are clearly visible. The following points remain to be addressed:

- Unit "years" is still lacking behind "20-79" in the Abstract, in the Results and in Figure 1. 

- Any results which are mentioned in the Abstract should also be mentioned in the main text (e.g. mean age).

- The interpretation that prediabetes was independently associated with preclinical atherosclerosis is very important for the main message of this paper and should therefore also be mentioned in the results section and explained how this conclusion was arrived at (i.e. that this association remained significant after adjustment for confounders). I would also suggest to move prediabetes above CV risk in table 3 to make this result more visible. It might also be worth to mention in the text how the OR of preclinical atherosclerosis by prediabetes would look like without adjustment for the confounders mentioned in table 3.

- It would be helpful to mention all the numbers from the flowchart also in the main text to explain to the Reader how the number of 183 patients who were excluded from further analysis for technical reasons was achieved.

- Figure 1: "n=7 subjects were exclused" should read "excluded", and "USG" be replaced by "US".

- Please mention which version of R was used.

- In all tables including the supplement, "C.I." should be replaced by "CI" for reasons of consistency.

- In table 3, the p-values should be mentioned to the right of the 95% CIs.

Further, as the supplementary figures and tables will probably not be quality-checked after acceptance, it is important that they fulfil the necessary standards right now. Please see specific comments below:

- Please add "Supplementary" before each figure and table.

- The supplementary tables should be ordered in the way they are mentioned in the text (i.e. supplementary table 1 should be placed between supplementary tables 4 and 5).

- Asteriks should be removed from the p-values, all p-values >0.01 should be presented with 2 decimals.

- Each table should start on a new page. 

- In supplementary table 2, the sentence "* there is a statistically significant difference after Bonferroni correction." should be replaced by "P-values were calculated using Bonferroni correction for the number of variables investigated (n=7)". 

- In supplementary tables 3 and 5, the sentence "Significance differences obtained by Chi2 test at the .05 level." should be replaced by "P-values were derived from Chi-square tests." and put into the figure legend (not in the footnote).

- Which test was applied in supplementary table 4?

- Combined columns such as in supplementary tables 3 (All participants with prediabetes) and 4 (Total) should be avoided. 

- Supplementary table 5 still contains coloured font. "p* Differences between population with or without prediabetes in a particular CVR category." should be removed. 

- Supplementary figure 2: This Figure might be removed altogether. If the authors decide to keep it, the following changes should be considered: The title "Model: Full" is not informative. "Logistic regression models" in the legend is confusing, as the ORs seem to have been derived from only one model. The values on the x-axis should be presented in horizontal direction. It should be mentioned that the values on the x-axis follow a logarithmic scale. Suggest to remove the OR of serum insulin level from this plot.

---

## [Author Response · Author response to Decision Letter 3]

12 Aug 2024

Additional Editor Comments:

The manuscript is now almost ready for journal acceptance. Please accept all changes made so far so that only changes required for this (hopefully last) revision are clearly visible. The following points remain to be addressed:

- Unit "years" is still lacking behind "20-79" in the Abstract, in the Results and in Figure 1. 

Done. 

- Any results which are mentioned in the Abstract should also be mentioned in the main text (e.g. mean age).

Done.

- The interpretation that prediabetes was independently associated with preclinical atherosclerosis is very important for the main message of this paper and should therefore also be mentioned in the results section and explained how this conclusion was arrived at (i.e. that this association remained significant after adjustment for confounders). I would also suggest to move prediabetes above CV risk in table 3 to make this result more visible. It might also be worth to mention in the text how the OR of preclinical atherosclerosis by prediabetes would look like without adjustment for the confounders mentioned in table 3.

Prediabetes was moved at the top of Table 3. We also added in the Results section:

Based on these results, we conclude that prediabetes was independently associated with preclinical atherosclerosis. This association remained significant after adjustment for confounders and the final findings were shown in Table 3.

- It would be helpful to mention all the numbers from the flowchart also in the main text to explain to the Reader how the number of 183 patients who were excluded from further analysis for technical reasons was achieved.

We have changed to “Additionally, 183 patients were excluded from further analysis for technical reasons – inability to perform an OGTT (n=132) or US of the carotid arteries (n=4) or lack of the HbA1c measurement (n=7) or lack of CVR assessment (n=40)”. 

- Figure 1: "n=7 subjects were exclused" should read "excluded", and "USG" be replaced by "US".

Done. 

- Please mention which version of R was used.

Done.

- In all tables including the supplement, "C.I." should be replaced by "CI" for reasons of consistency.

Done.

- In table 3, the p-values should be mentioned to the right of the 95% CIs.

Done.

Further, as the supplementary figures and tables will probably not be quality-checked after acceptance, it is important that they fulfil the necessary standards right now. Please see specific comments below:

- Please add "Supplementary" before each figure and table.

Done.

- The supplementary tables should be ordered in the way they are mentioned in the text (i.e. supplementary table 1 should be placed between supplementary tables 4 and 5).

Changed. 

- Asteriks should be removed from the p-values, all p-values >0.01 should be presented with 2 decimals.

Done 

- Each table should start on a new page. 

Done

- In supplementary table 2, the sentence "* there is a statistically significant difference after Bonferroni correction." should be replaced by "P-values were calculated using Bonferroni correction for the number of variables investigated (n=7)". 

Done.

- In supplementary tables 3 and 5, the sentence "Significance differences obtained by Chi2 test at the .05 level." should be replaced by "P-values were derived from Chi-square tests." and put into the figure legend (not in the footnote).

Done.

- Which test was applied in supplementary table 4?

Chi2 test was applied. I added in the caption of Supplementary table 4. 

- Combined columns such as in supplementary tables 3 (All participants with prediabetes) and 4 (Total) should be avoided. 

Done.

- Supplementary table 5 still contains coloured font. "p* Differences between population with or without prediabetes in a particular CVR category." should be removed. 

Done. 

- Supplementary figure 2: This Figure might be removed altogether. If the authors decide to keep it, the following changes should be considered: The title "Model: Full" is not informative. "Logistic regression models" in the legend is confusing, as the ORs seem to have been derived from only one model. The values on the x-axis should be presented in horizontal direction. It should be mentioned that the values on the x-axis follow a logarithmic scale. Suggest to remove the OR of serum insulin level from this plot.

Thank you. We decided to remove this figure.

---

## [Editor Report · Decision Letter 4]

21 Aug 2024

The impact of prediabetes on preclinical atherosclerosis in general apparently healthy population: a cross-sectional study

PONE-D-23-42492R4

Dear Dr. Kaminski,

We’re pleased to inform you that your manuscript has been judged scientifically suitable for publication and will be formally accepted for publication once it meets all outstanding technical requirements.

Kind regards,

Andreas Beyerlein

Academic Editor

PLOS ONE
---

## [Editor Report · Acceptance letter]

28 Aug 2024

PONE-D-23-42492R4 

PLOS ONE

Dear Dr. Kamiński , 

I'm pleased to inform you that your manuscript has been deemed suitable for publication in PLOS ONE. Congratulations! Your manuscript is now being handed over to our production team.

Kind regards, 

on behalf of

Dr. Andreas Beyerlein 

Academic Editor

PLOS ONE